# EXPLORING EXPERT CONCENTRATION FOR PARAMETER-EFFICIENT FINE-TUNING OF MIXTURE-OF-EXPERT LLMS

## ABSTRACT

Scaling large language models (LLMs) with the Mixture-of-Experts (MoE) architecture has emerged as a powerful alternative to dense models. However, fine-tuning MoE models for domain- or task-specific adaptation remains challenging: full-model tuning is prohibitively expensive, while existing parameter-efficient fine-tuning (PEFT) methods, mostly adapted from dense models, suffer from unstable optimization due to MoE's sparse expert activation. In this work, we conduct an empirical study on the fine-tuning dynamics of MoE models. We first introduce the Domain Advantage Score (DAS), a simple yet effective metric for identifying domain-relevant experts. Our findings uncover an expert concentration phenomenon: during domain-specific fine-tuning, the overall DAS of the top experts consistently increases, indicating a progressive enhancement of domain concentration. Building on this, we propose a lightweight two-stage PEFT framework: (1) fine-tuning only the attention and router layers to sharpen expert specialization, and (2) selectively fine-tuning parameters on the identified experts. This approach updates only a small fraction of parameters while achieving performance on par with full fine-tuning, and it effectively preserves the model's general capabilities. Experiments on nine benchmarks show the effectiveness and efficiency of our method. Our code and data will be publicly released.

## 1 INTRODUCTION

Scaling laws demonstrate that model performance improves predictably with increasing parameters, making parameter scaling a central driver of progress in large language models. While dense architectures have delivered strong results, their computational and memory demands grow prohibitively at large scales. To address this, the Mixture-of-Experts (MoE) architecture (Shazeer et al., 2017; Zhou et al., 2022; Dai et al., 2024) has become a dominant paradigm for scaling beyond dense models. MoE organizes the model into a large pool of experts but activates only a small subset of them for each token during inference, enabling sparse activation that dramatically improves efficiency while retaining capacity. This design allows MoE models to reach billions of parameters without linearly increasing inference cost, and they have already achieved remarkable performance across a range of tasks, establishing MoE as a cornerstone architecture for next-generation LLMs.

Fine-tuning or continual pre-training Mixture-of-Experts (MoE) models on specific domains or tasks is crucial for adapting to real-world applications. However, the massive parameters of MoE models makes full-model tuning prohibitively expensive. To mitigate this, researchers have attempted to transfer parameter-efficient fine-tuning (PEFT) techniques originally developed for dense models (e.g., adapters (Houlsby et al., 2019) and LoRA (Hu et al., 2022)) to the MoE setting (Zadouri et al., 2024; Dou et al., 2024; Liu et al., 2024b). Despite the reduced cost, they often struggle to match the effectiveness achieved in dense models, because MoE's sparse activation introduces unique challenges. Specifically, only a small subset of experts is activated for each token, which leads to unstable gradient flow and hampers optimization during fine-tuning (Guo et al., 2025).

To better understand how to perform parameter-efficient fine-tuning (PEFT) for Mixture-of-Experts (MoE) models, we first study the fine-tuning dynamics. We introduce the Domain Advantage Score (DAS), defined as the difference between an expert's selection frequency on the target domain and

its frequency on a general dataset, to quantify the affinity of an expert to a specific domain. Our analysis reveals a phenomenon of expert concentration: when an MoE is fine-tuned on domain- or task-specific data, the cumulative DAS of top-ranked experts increases, indicating domain-specific experts more distinct and easier to identify. Building on this finding, we propose a simple metric to identify task- or domain-relevant experts before fine-tuning. By restricting fine-tuning to these selected experts, we achieve performance comparable to full expert tuning, while also reducing catastrophic forgetting and better preserving general capabilities.

Building on the observed phenomenon of expert concentration, we propose a lightweight two-stage tuning framework for MoE models that further reduces the number of trainable parameters. In the first stage, we perform Attention and Router Tuning, updating only the attention and router layers (around 2.5% of total parameters) while keeping all experts frozen. This stage exploits the natural increase in routing scores during fine-tuning, which sharpens the concentration of experts and makes domain-relevant ones more distinguishable. In the second stage, we apply our proposed metric to identify the most specialized experts and only require to fine-tune these experts. This design achieves efficient adaptation to new domains by combining routing-driven concentration with selective expert tuning, reaching performance comparable to full fine-tuning, by training totally 8% parameters.

We evaluate our method on multiple math and coding benchmarks and demonstrate its superiority than other parameter-efficient fine-tuning methods. Besides, the stable performance on general benchmarks also indicates the effectiveness of our method on resisting catastrophe forgetting.

The main contributions of this work are as follows:

• We uncover an expert concentration phenomenon in MoE fine-tuning, indicating stronger domain alignment and clearer separation between domain-aligned and general experts.

• Based on this finding, we design a simple metric to identify task-relevant experts, enabling selective fine-tuning that matches full-model performance while reducing catastrophic forgetting.

• Building on this, we propose a lightweight two-stage PEFT framework that first tunes attention and routers, then selectively fine-tunes expert modules, achieving near full-tuning accuracy with only a small fraction of parameters.

• Extensive experiments on specific domain data and general benchmarks have shown the effectiveness of our methods in achieving good performance and resisting catastrophe forgetting.

## 2 EMPIRICAL STUDY

To design effective parameter-efficient fine-tuning strategies for Mixture-of-Experts (MoE) models, it is crucial to first understand their fine-tuning dynamics. In this section, we empirically analyze how expert routing distributions evolve during domain adaptation and investigate whether fine-tuning needs to involve all experts or only a subset.

### 2.1 EXPERT CONCENTRATION PHENOMENON

We first investigate the dynamics of domain expert routing during fine-tuning. To quantify the affinity of experts to a specific domain, we introduce the Domain Advantage Score (DAS), a metric designed to measure how strongly each expert specializes in a target domain. For an expert, its DAS for the domain-specific data $\mathcal{D}_d$ and general data $\mathcal{D}_g$ is computed as

$$\mathrm{DAS}(\mathcal{D}_d, \mathcal{D}_g) = \frac{1}{|\mathcal{D}_d|} \sum_{t \in \mathcal{D}_d} g_t - \frac{1}{|\mathcal{D}_g|} \sum_{t \in \mathcal{D}_g} g_t, \tag{1}$$

where $g_t$ is the routing score of the expert for token $t$. A larger DAS indicates stronger domain affinity, distinguishing domain-specific experts from others. Besides, to quantify how strongly domain advantage concentrates on head experts, we use Top-k Cumulative Domain Advantage (C-DAS@k):

$$\mathrm{C\text{-}DAS}@k = \sum_{i=1}^{L} \frac{\sum_{j \in \mathcal{T}_i} \max(\mathrm{DAS}_{ij}, 0)}{\sum_{j=1}^{N} \max(\mathrm{DAS}_{ij}, 0)} \tag{2}$$

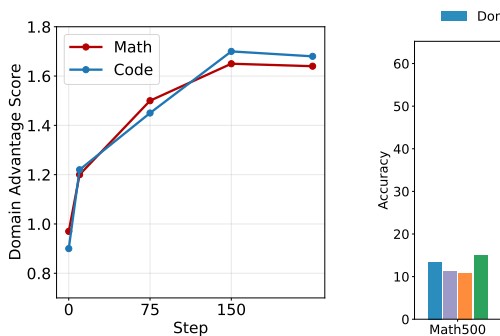 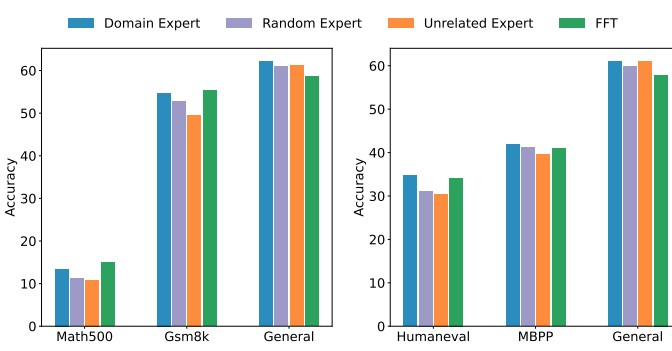

(a) The Dynamics of C-DAS@$k$.      (b) Effect of Training Different Types of Expert.

Figure 1: Results from the empirical study: (a) the curve of increasing Top-k Cumulative Domain Advantage (C-DAS@6) with respect to the training steps during fine-tuning; (b) performance comparison of fine-tuning different subsets of experts.

where $\text{DAS}_{ij}$ denotes the DAS of the $j$-th expert in the $i$-th layer. $T_i$ denotes the indices of the Top-k experts at layer $i$ ranked by C-DAS@$k$. A higher C-DAS@$k$ indicates a more pronounced and specialized functionality of the expert for the given special domain data.

**Experimental Setup.** We fine-tune MoE language models DeepSeek-V2-Lite (DeepSeek-AI et al., 2024) on two domain datasets, i.e., mathematics and programming code. Concretely, we select MATH500 (Hendrycks et al., 2021b) and GSM8K (Cobbe et al., 2021) to assess mathematical reasoning, while HumanEval (Chen et al., 2021) and MBPP (Austin et al., 2021) are selected to measure coding ability. To better analyze how the routing scores changed, we freeze the expert feed-forward blocks and update only the attention and router layers. During training, we track the evolution of the Cumulative DAS for the top-6 experts(10% of the total experts) to examine how domain advantage concentrates among the top experts.

**Finding-1: Fine-tuning concentrates domain advantage into a small head set of experts.** As shown in Figure 1a, the model's C-DAS@6 steadily increases over training, indicating the concentrated domain advantage on top experts. In effect, routing becomes more selective, and tokens from the target domain are progressively steered toward a few experts whose domain affinity grows, sharpening the separation between domain-aligned and generalist experts. We also observe that the top-$k$ ranking stabilizes early, meaning the same small subset repeatedly captures most of the positive DAS. These findings support two conclusions: (i) fine-tuning primarily strengthens the already relevant experts instead of uplifting all experts uniformly, and (ii) a small, stable set of high-DAS experts suffices for adaptation. This directly motivates our PEFT design: first use Attention and Router updates to expose domain-aligned experts, then selectively fine-tune only the identified high-DAS experts to capture most of the in-domain gains while minimizing interference and forgetting.

## 2.2 CONCENTRATED EXPERT FINE-TUNING

Building on DAS, we empirically explore the impact of fine-tuning different types of experts for domain-specific tasks, we construct three distinct expert subsets for fine-tuning:

- Domain Experts: the experts with the highest DAS values, reflecting strong domain specialization.

- Random Experts: experts sampled uniformly at random, serving as a baseline.

- Unrelated Experts: those with the lowest DAS values, least aligned with the target domain.

This formulation ensures that expert selection is based on true domain preference learned from training, enabling us to test whether focusing on specialized experts suffices for effective fine-tuning.

**Finding-2: fine-tuning head experts lead to better performance in domain and general tasks.** As shown in Figure 1b, fine-tuning Domain Experts consistently outperforms the other two subsets,

Random Experts yield moderate gains, and Irrelevant Experts result in worse performance. DAS-ranked Domain Experts already attract in-domain routing traffic, so their updates align with the dominant gradient signal, improving sample efficiency and accelerating convergence. By contrast, updating Irrelevant Experts diverts capacity away from the active pathways, and injects gradient noise into experts that see little in-domain usage, which degrades the target-domain accuracy. Besides, DAS-selected top experts preserves general capabilities better than full or random fine-tuning, because it minimizes interference on non-specialized experts. Together, these findings confirm that MoE models can be efficiently adapted by focusing updates on a small DAS-identified expert subset while reducing compute and mitigating catastrophic forgetting.

## 3 METHOD

Motivated by the observation of the expert concentration phenomenon, we aim to propose a more efficient fine-tuning method for MoE LLMs. Since domain-specific fine-tuning naturally concentrates on a small set of experts (Dong et al., 2025), we first frozen all the experts and only fine-tune the attention and routing layers until convergence, to help identify the concentrated experts. Then, we fine-tune only the parameters in the concentrated few experts identified by the DAS. The whole process totally fine-tunes average 8% parameters, and the two-stage localized training paradigm can alleviate the unstable optimization and catastrophe forgetting issues.

### 3.1 PRELIMINARIES: MIXTURE-OF-EXPERTS

The Mixture-of-Experts (MoE) framework (Jacobs et al., 1991; Jordan & Jacobs, 1994) scales model capacity by partitioning computation across multiple experts. An MoE layer consists of $N$ experts $\{E_i\}_{i=1}^N$ and a router $R$. Given an input token $\mathbf{x}^{(l)}$ at layer $l$, the router computes a routing value vector $\mathbf{g}$, and only top-$k$ experts with the highest routing values are aggregated to the hidden state:

$$\mathbf{g} = \text{softmax}(R(\mathbf{x}^{(l)}, \theta_R)); \ \mathbf{h}^{(l)} = \sum_{i \in \text{top-k}(\mathbf{g})} g_i \cdot E_i(\mathbf{x}^{(l)}). \tag{3}$$

This sparse activation enables MoE models to scale to billions of parameters with sublinear inference cost. However, the same sparsity complicates fine-tuning, as only a small subset of experts are consistently updated, leading to instability and inefficiency.

### 3.2 PARAMETER-EFFICIENT MoE FINE-TUNING

Our proposed efficient MoE fine-tuning strategy consists of two stages, i.e., attention and router fine-tuning and DAS-guided experts fine-tuning. The overall framework is illustrated in Figure 2.

**Stage 1: Attention and Router Fine-tuning.** In the first stage, we freeze all expert feed-forward networks (FFNs) and embedding layer, and update only the attention layers and router modules. These components account for roughly 2.5% of the total parameters, making this stage lightweight yet highly effective. Since the attention layer and router determine expert assignment and token routing traffic, according to Finding-1 in Section 2.1, tuning them allows the model to gradually concentrate routing probabilities on a small subset of domain-relevant experts. This

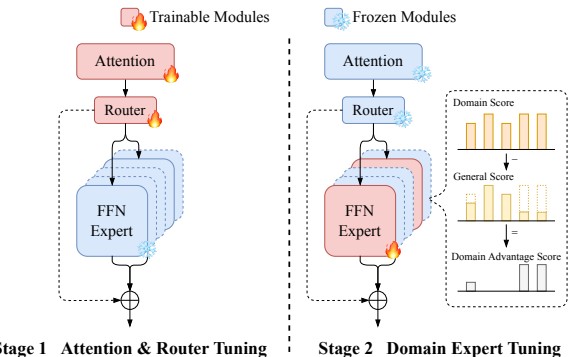

Figure 2: Overview of our DAS-guided two-stage fine-tuning framework. Stage 1 tunes attention and router modules, while stage 2 ranks experts by DAS and fine-tunes only on the top-ranked experts.

sharpening process not only clarifies which experts specialize in the target domain but also avoids the instability that arises when all experts are updated simultaneously. By the end of Stage 1, the

model develops a clearer separation between domain-specialized experts and general-purpose ones, which can be systematically quantified through our Domain Advantage Score (DAS).

**Stage 2: DAS-guided Expert Fine-tuning.** After stage-1, the trained router and attention layers can make the domain-relevant experts more outstanding, which are easy to be identified by our proposed DAS values. Specifically, we compute DAS values across all experts to rank their domain affinity and retain only the top $k$ experts. Then, we move to the second stage, which only requires to train the parameters within the top-ranked experts. Here, we can choose to train all the parameters of these experts (about 8% parameters) or the LoRA adapters on them (about 1% parameters). As we keep the majority of the network frozen, both settings ensure efficient adaptation and lower training cost. Crucially, because the router distribution has already been aligned in stage-1, these selected experts now capture domain knowledge more effectively, mitigating catastrophic forgetting and preserving general abilities on out-of-domain tasks.

# 4 EXPERIMENTS

## 4.1 EXPERIMENTAL SETUP

**MoE Models.** We evaluate our approach on three widely used opensource MoE-based LLMs for evaluation: DeepSeek-V2-Lite (DeepSeek-AI et al., 2024), DeepSeek-MoE-Base (Dai et al., 2024) and Qwen1.5-MoE-A2.7B (Yang et al., 2025). These models provide complementary architectures to assess the robustness and generality of our method. To ensure comparability, all experiments are conducted using greedy decoding, which yields consistent and deterministic outputs across models.

**Dataset.** We conduct evaluations on three categories of benchmarks designed to assess mathematical, coding and general abilities. To ensure alignment between supervision stage and downstream evaluation, the training and test sets are organized according to related domains.

- **Mathmatical reasoning ability**: we regenerate solutions for MetaMathQA (Yu et al., 2024) and retain only verified-correct chain-of-thought traces as supervision, and report evaluation results on GSM8K (Cobbe et al., 2021) and MATH-500 (Hendrycks et al., 2021b);
- **Coding ability**: we fine-tune on a filtered subset of the OpenCoder corpus (Huang et al., 2025) and evaluate performance on HumanEval (Chen et al., 2021) and MBPP (Austin et al., 2021).
- **General ability**: To gauge trade-offs in general capability after domain-targeted finetuning, we evaluate on CommonsenseQA (Talmor et al., 2019), ARC-Challenge (Clark et al., 2018),StrategyQA (Geva et al., 2021), CEval (Huang et al., 2023) and MMLU (Hendrycks et al., 2021a), covering natural-language understanding and commonsense QA beyond the training domains.

**Baseline Methods.** We compare our method with four MoE fine-tuning strategies: Fully Fine-Tuning (FFT), LoRA (Hu et al., 2022) and Expert-Specialized(ESFT) (Wang et al., 2024). ESFT leverages expert specialization by updating only a pre-selected subset of experts for a target task, while leaving the router frozen. As the subset is identified from the router's routing distribution, MoE load-balancing constraints may bias selection toward capacity considerations rather than task alignment, potentially yielding suboptimal expert choices.

**Implementation Details.** All experiments use a batch size of 8 and a sequence length of 1,024. For each task, training is capped at 1,000 steps with evaluation every 50 steps. we set learning rate1e-4 for LoRA and 5e-5 for all other methods based on a hyperparameter search in {1e-5, 2e-5, 5e-5, 1e-4}. LoRA uses rank 16 with lora_alpha = 32.

## 4.2 MAIN RESULTS

Table 1 summarizes the experimental results. Under the same training budget, our method achieves the best accuracy across all evaluated reasoning benchmarks and for each of the three MoE backbones. The improvements are consistent, not tied to a particular architecture or dataset, which suggests that the proposed adaptation pathway generalizes well. We attribute these gains to the two-stage design: (i) first aligning routing so tokens of different types are dispatched to the most

| Model | Method | Para. | GSM8K | MATH | MBPP | Humaneval | Avg. |
|---|---|---|---|---|---|---|---|
| | - | | 43.38 | 10.80 | 40.80 | 30.48 | 35.45 |
| | FFT | 100% | 55.34 | 15.00 | 42.60 | 34.15 | 43.25 |
| Deepseek-V2-Lite | LoRA | 2% | 51.10 | 13.00 | 39.40 | 29.87 | 39.66 |
| | ESFT | 8% | 52.46 | 13.20 | 39.00 | 32.92 | 40.55 |
| | DAS-Tune | 8% | **54.73** | **13.40** | **42.60** | **34.75** | **42.64** |
| | DAS-LoRA | ≤ 1% | 52.00 | 13.00 | 39.40 | 29.87 | 40.14 |
| | - | | 18.80 | 3.80 | 39.20 | 26.21 | 20.37 |
| | FFT | 100% | 37.90 | 7.20 | 42.60 | 33.54 | 32.37 |
| Deepseek-MoE-Base | LoRA | 2% | 27.44 | **6.00** | 38.80 | 28.04 | 25.44 |
| | ESFT | 8% | 32.14 | 5.20 | 39.20 | 28.65 | 27.90 |
| | DAS-Tune | 8% | **33.81** | 5.40 | **40.40** | **29.87** | **29.15** |
| | DAS-LoRA | ≤ 1% | 31.61 | 5.00 | 39.20 | **29.87** | 27.66 |
| | - | | 61.33 | 15.20 | 42.8 | 34.20 | 46.52 |
| | FFT | 100% | 67.43 | 19.15 | 44.00 | 38.54 | 51.08 |
| Qwen-MoE-A2.7B | LoRA | 2% | 65.13 | 15.50 | 42.00 | 36.50 | 48.58 |
| | ESFT | 8% | 65.13 | 16.20 | 43.20 | 36.80 | 48.98 |
| | DAS-Tune | 8% | **65.57** | **17.20** | **43.60** | **37.15** | **49.52** |
| | DAS-LoRA | ≤ 1% | 64.37 | 16.00 | 42.40 | 36.80 | 48.38 |

Table 1: Experimental results across different fine-tuning methods and tasks on three MoE backbones. Para. denotes the trainable parameter percentage in the model. Avg. is the average value of all categories. The best results among all non-FFT methods are denoted in bold.

| | CSQA | ARC-C | StrategyQA | CEval | MMLU | Avg. |
|---|---|---|---|---|---|---|
| DeepSeek-V2-Lite | 61.34 | 63.37 | 55.74 | 59.82 | 57.50 | 60.36 |
| +LoRA | 60.94 | 61.26 | 55.26 | 58.20 | 56.42 | 59.26 |
| +FFT | 58.61 | 59.47 | **56.04** | 57.92 | 55.50 | 57.96 |
| +ESFT | 61.26 | **63.97** | 53.65 | **60.05** | 57.00 | 60.08 |
| +Ours | **61.99** | 62.97 | 54.89 | 60.05 | **57.30** | **60.27** |

Table 2: Experimental results on general tasks to test the general ability degradation after fine-tuning. We add the backbone performance as reference, and the best methods are denoted in bold.

suitable experts, and (ii) then refining only the small expert subset most relevant to the target tasks. This sequence reduces gradient interference, sharpens domain specialization, and yields stronger task alignment without inflating the update cost.

In terms of efficiency, our approach updates roughly 8% of parameters while reaching performance close to full fine-tuning (FFT), amounting to an $12\times$ reduction in the number of trainable weights. Within a fixed step and data budget, this produces near-FFT accuracy at a fraction of the compute and memory footprint, highlighting a practical route to adapt large MoE models when resources are constrained. Although vanilla LoRA provides the smallest storage overhead, its downstream performance trails other methods in the sparse MoE setting, indicating that minimizing parameter count alone is insufficient when expert routing and specialization dynamics are central to transfer.

Table 2 reports general-ability evaluations. Our method exhibits the smallest degradation relative to all baselines, indicating better retention of pre-existing capabilities after domain adaptation. We believe this stability stems from avoiding indiscriminate updates: full or broadly targeted expert tuning can erode established specializations and perturb load balancing, whereas our DAS-guided selection confines updates to the experts already aligned with the target domain. As a result, the adapted models maintain broader competency while still delivering strong, domain-specific gains.

## 4.3 FURTHER ANALYSIS

Following our main experiments, we conduct detailed analysis experiments to demonstrate the effectiveness of our method and to explore the characteristics of identified domain experts. Unless specified, all analysis results are based on the DeepSeek-V2-Lite model.

| Task | Before | After | RPR |
|------|--------|-------|-----|
| GSM8K | 54.73 | 53.53 | 0.978 |
| MATH | 13.40 | 13.00 | 0.970 |
| Avg. | 43.36 | 42.38 | 0.977 |

(a) Phase-1 **Math** $\Rightarrow$ Phase-2 **Code**

| Task | Before | After | RPR |
|------|--------|-------|-----|
| HumanEval | 34.75 | 34.14 | 0.982 |
| MBPP | 42.60 | 42.20 | 0.990 |
| Avg. | 40.66 | 40.21 | 0.988 |

(b) Phase-1 **Code** $\Rightarrow$ Phase-2 **Math**

Table 3: Ability retention study for continual training MoE using our method on new domains. We report the retained performance ratio $\mathrm{RPR} = \text{After}/\text{Before}$.

| | GSM8K | MATH500 | MBPP | humaneval | Avg. |
|---|-------|---------|------|-----------|------|
| Ours | **54.73** | **13.40** | **42.60** | **34.75** | **42.64** |
| - Attention Tuning Only | 53.37 | 12.80 | 41.00 | 33.53 | 41.39 |
| - FFN Tuning Only | 51.48 | 10.80 | 39.6 | 32.31 | 39.62 |
| Backbone | 43.38 | 10.80 | 40.80 | 30.48 | 35.45 |

Table 4: Ablation Study Results on DeepSeek-V2-Lite. All experiments were run for the same total training steps to ensure a fair comparison. The best results are denoted in bold.

**Continual Learning Study.** To investigate whether our method causes catastrophic forgetting, we designed a simple but revealing continued fine-tuning experiment. Specifically, for a model that has been fine-tuned on a coding dataset, we apply our two-stage method to fine-tune it on a mathematics dataset for an equal number of steps. We perform the same experiment in reverse, fine-tuning a math-trained model on a coding dataset. By measuring the model's performance on its original domain before and after the secondary fine-tuning, we can assess the extent of knowledge degradation. As shown in Table 3, our method effectively preserves the model's original knowledge. Despite continued fine-tuning on a different domain, the model's performance on its original task remains largely stable, with only a negligible drop. This demonstrates that our approach, by selectively updating only the most domain-relevant parameters, avoids damaging the model's foundational knowledge.

**Ablation Study.** To validate the efficacy of our proposed two-stage fine-tuning approach, we conduct an ablation study comparing it against two single-stage baselines, all with an identical total number of training steps (1000 steps) to ensures a fair comparison of their respective strategies. The baselines are: 1) Attention-Tuning Only, where we exclusively fine-tune the Attention and Router modules for all 1000 steps; and 2) FFN-Tuning Only, where we fine-tune the expert FFNs for all 1000 steps, with the expert subset selected based on their pre-tuning DAS. As shown in Table 4, our two-stage method consistently achieves the best average performance across all datasets. We attribute this superior performance to the unique synergy between the two stages. The initial Attention-Tuning phase dynamically refines the expert distribution, acting as a powerful pre-selector that optimizes the expert subset for the subsequent stage. This allows the second FFN-Tuning phase to apply computational resources precisely to the most relevant and specialized experts, leading to a more substantial performance gain. This result demonstrates that simply training a specific component or a pre-selected expert group is suboptimal, and that the two-stage adaptive process is crucial for achieving peak performance with MoE fine-tuning.

**Variation Study of Expert Identification Method.** To validate the effectiveness of the Domain Advantage Score (DAS), we conduct an ablation study comparing it against several alternative strategies for identifying domain-relevant experts. We evaluate each method by using its top-ranked experts for fine-tuning and measuring the resulting performance on a target domain. The alternative methods explored are: (1) Direct Routing Score: The average gate score of an expert on the domain dataset; (2) Expert Output Norm: The average L2 norm of an expert's output on the domain dataset; (3) Expert Contribution: The contribution of an expert to the change in the hidden state, reflecting its impact on the model's output; (4) Product of Score and Norm: The average product of an expert's gate score and its output norm. As shown in Table 5, expert selection guided by the Domain Advantage Score (DAS) consistently outperforms all alternative methods on the downstream task. We attribute this to DAS's relative nature: by contrasting domain and general routing mass, it focuses updates on experts whose activations are selectively elevated by the target domain.

|  | GSM8K | MATH500 | MBPP | humaneval | Avg. |
|---|---|---|---|---|---|
| Ours (using DAS) | **54.73** | 13.40 | **42.80** | **34.75** | **42.69** |
| - Score | 52.46 | 13.40 | **42.80** | 34.14 | 41.43 |
| - ExpLen | 52.91 | 12.40 | 42.20 | 34.75 | 41.39 |
| - EC | 53.52 | 13.00 | 41.80 | 33.53 | 41.68 |
| - PSN | 54.05 | **13.60** | **42.80** | 34.14 | 42.32 |
| Backbone | 43.38 | 10.80 | 40.80 | 30.48 | 35.45 |

Table 5: Impact of Expert Identification Methods on Fine-Tuning Performance. ExpLen, EC, PSN and DAS denote Average Expert Output Norm, Expert Contribution, Product of Score and Norm and Domain Advantage Score. The best results are denoted in bold.

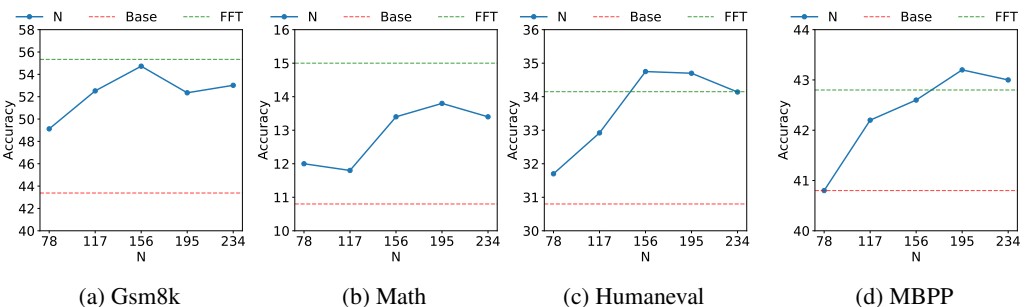

(a) Gsm8k      (b) Math      (c) Humaneval      (d) MBPP

Figure 3: Comparison of our method with varying numbers of trainable experts (N) against the Base model and Full Fine-Tuning (FFT) results.

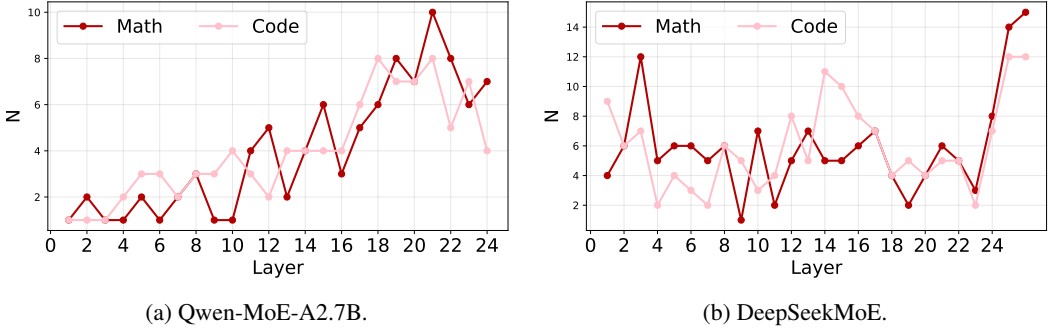

(a) Qwen-MoE-A2.7B.      (b) DeepSeekMoE.

Figure 4: Distribution of domain experts across layers identified by DAS.

**Effect of Trainable Expert Count.** To quantify the effect of expert subset size on performance, we vary the number of fine-tuned experts identified by our first-stage, attention-guided procedure from 78—approximately 4% of total parameters—up to 234—approximately 12%—under a fixed compute budget with identical tokens, steps, and optimizer settings. We compare these variants against greedy decoding without fine-tuning as well as full fine-tuning. As shown in Figure 3, accuracy rises as the subset grows from very small budgets to about 10% of parameters, after which additional experts deliver diminishing returns within the same training horizon. Beyond this knee point, the gap to full fine-tuning narrows only slightly, indicating that most task-relevant routing mass has already been captured and that enlarging the updated subset disperses gradients over low-traffic experts, thereby reducing update efficiency. Overall, a compact expert set around 8–10% of parameters recovers the majority of attainable gains under limited steps, and coordinated, router-aware selection proves more consequential than indiscriminately expanding the fine-tuned subset.

**Domain-Specific Expert Distribution.** Furthermore, we analyze the distribution of domain-specific experts identified by our method and report counts per layer for two MoE backbones in Figure 4a and Figure 4b. Across domains, the selected experts concentrate toward the final layers, while the middle portion of the network contains fewer domain experts. This profile indicates that

middle layers exhibit higher selectivity and lower coverage, consistent with a more peaked routing pattern that relies on a small set of broadly useful experts, whereas deeper layers host a richer pool of domain-specialized experts. These observations suggest a depth-progressive organization of knowledge: early and middle layers prioritize generic transformations that transfer across domains, and deeper layers encode domain-specific mechanisms that benefit most from targeted adaptation.

## 5 RELATED WORK

**Parameter-efficient Fine-tuning for Transformers.**    As Transformer models continue to grow in scale, full fine-tuning (Qiu et al., 2020) has become increasingly impractical. parameter-efficient fine-tuning (PEFT) mitigates this by updating a small subset or a low-rank reparameterization of weights. Representative families include adapter tuning (Houlsby et al., 2019; Sung et al., 2022), prompt tuning (Lester et al., 2021), and reparameterized low-rank updates such as LoRA (Hu et al., 2022) and its variants (e.g., DoRA (Liu et al., 2024a)) that improve stability or capacity. Notably, all these methods primarily focus on adapting dense models, leaving the application of PEFT to inherently sparse Mixture-of-Experts (MoE) models comparatively underexplored. While parameter-efficient fine-tuning PEFT has matured for dense Transformers, its application to inherently MoE architectures remains comparatively underexplored. One line of MoE-tuning work integrates adapter-style or low-rank updates directly into MoE components and coordinates them with the router so that adaptation follows the model's sparse computation (Liu et al., 2024c). Another leverages expert specialization by selectively fine-tuning a small, task-relevant subset of experts while freezing the rest (Wang et al., 2024). In both cases, parameter updates are confined to lightweight subblocks, e.g., the feed-forward (FFN) or attention modules, treating attention and experts in isolation or relying on static expert selection, which can misalign routing context with expert updates.

**Sparsity and Specialization in MoE Architectures.**    Unlike dense models where all parameters are activated for every token, MoE (Shazeer et al., 2017; Zhou et al., 2022) routes tokens to a small subset of "expert" sub-networks. This sparse activation mechanism allows for a significant increase in model size without a proportional increase in computational cost during inference. Recent advances in Mixture-of-Experts architectures have explored both coarse-grained (Jiang et al., 2024) and fine-grained expert paradigms (Dai et al., 2024; Yang et al., 2025). In early models, the number of experts was often limited, with coarse-grained routing activating a small, fixed subset. More recent research, however, has increasingly focused on fine-grained MoE designs where a much larger pool of experts is available, but only a few are sparsely activated per token. Empirical studies consistently show that fine-grained configurations exhibit a high degree of expert specialization (DeepSeek-AI et al., 2024; Lu et al., 2024): domain traffic concentrates on a compact subset of experts (Dong et al., 2025). As a result, identifying non-domain experts via domain data and pruning or masking them tends to have minor impact on downstream domain performance (Muzio et al., 2024; Xie et al., 2024; He et al., 2024), indicating a structured, overcomplete form of specialization in sparse MoE. This inherent specialization provides a pathway for efficient fine-tuning. By identifying and selecting a small subset of task-relevant experts, the computational cost of adapting a massive MoE model to a new task can be significantly reduced.

## 6 CONCLUSION

In this paper, we investigated the fine-tuning dynamics of Mixture-of-Experts (MoE) models and revealed the expert concentration phenomenon, where experts' relative domain specialization is progressively amplified during domain-specific adaptation. This finding indicates that full-model fine-tuning is not only costly but also unnecessary, since a few domain-relevant experts capture the majority of task knowledge. To systematically identify these experts, we introduced the Domain Advantage Score (DAS), which quantifies domain affinity by contrasting expert routing behaviors on domain versus general data. Building on this insight, we proposed a lightweight two-stage parameter-efficient tuning framework: first aligning routing signals through attention and router tuning, and then selectively fine-tuning parameters of DAS-identified experts. Extensive experiments on math and coding benchmarks demonstrate that our approach achieves performance comparable to full fine-tuning while updating only a small fraction of parameters, and it also mitigates catastrophic forgetting on general benchmarks.

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

## USE OF LARGE LANGUAGE MODELS

This manuscript used large language models in a narrowly circumscribed role: copy-editing for grammar and readability and occasional, non-substantive debugging hints. No model contributed to conceptual design, algorithmic choices, experiment execution, analysis, or claims. All technical content was authored, verified, and is fully owned by the authors.

## A  EFFECT OF COMPONENT TUNING ON MOE EXPERT ROUTING DYNAMICS

To quantify how much the average routing distribution shifts from the pre-tuning to the post-tuning model on domain data, we propose a metric called Routing Consistency(RC). For each expert, let $g_{ij}^{(1)}$ be the average routing score of the $j$-th expert in the $i$-th layer before fine-tuning, and $g_{ij}^{(2)}$ be the average routing score after fine-tuning. The shift for each expert is calculated as the squared L2-norm of the difference:

$$shift_{ij} = ||g_{ij}^{(2)} - g_{ij}^{(1)}||^2 \tag{4}$$

The overall Distribution Shift for the entire model is defined as the average shift across all layers and all experts:

$$RC = \frac{1}{L \times N} \sum_{i=1}^{L} \sum_{j=1}^{N} shift_{ij} \tag{5}$$

A lower Distribution Shift value indicates that the routing distribution has undergone minimal change.

We begin by computing the initial Domain Advantage Score (DAS) to identify domain-related experts, and then design two controlled interventions to disentangle how different modules affect routing. In the first intervention we fine-tune only the expert blocks while freezing attention and the router, in the second we fine-tune only the attention and router while freezing all experts. As shown in Figure 5, the Routing Consistency (RC) remains near its pre-tuning level under FFN-only updates, whereas RC shift significantly when updating attention&router.This indicates that FFN updates primarily change what an expert computes, leaving token-to-expert assignment largely intact, while attention&router directly reshape how token-level evidence is aggregated and converted into routing logits, thereby realigning the allocation of domain traffic across experts.

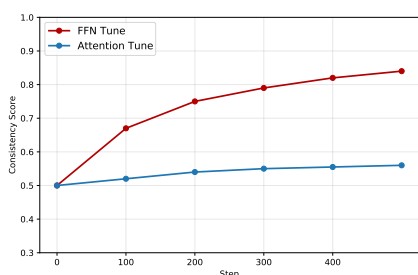

Figure 5: The Dynamics of RC.

## B  IMPACT OF ATTENTION&ROUTER-TUNING STEPS ON PERFORMANCE

To determine the optimal duration of our first-stage fine-tuning, we conducted an analysis on how the number of Attention-Tuning steps affects overall performance. By keeping all other variables constant, we varied the number of steps in the first stage from 100 to 500 and observed the impact on the downstream task.

As shown in Table 6, we found that increasing the number of attention-tuning steps generally improves performance. However, the performance gains exhibit diminishing returns after a certain point. This suggests that a limited number of steps in the first stage is sufficient to effectively steer the router and amplify the specialization of domain-relevant experts. Beyond this, additional steps do not yield a proportional increase in performance.

| Tuning Steps | GSM8K | MATH500 | MBPP | humaneval | Avg. |
|---|---|---|---|---|---|
| 100 | 53.37 | 12.00 | 39.80 | 29.87 | 40.75 |
| 200 | 53.52 | 12.80 | 41.40 | 31.10 | 41.39 |
| 300 | 53.98 | 13.40 | 41.20 | 32.92 | 41.84 |
| 400 | 54.73 | 13.80 | 42.20 | 32.31 | 42.48 |
| 500 | 54.73 | 13.40 | 42.60 | 34.75 | 42.64 |
| Backbone | 43.38 | 10.80 | 40.80 | 30.48 | 35.45 |

Table 6: Impact of Attention-Tuning Steps on Two-Stage Performance.

