# OpenReview forum: "Exploring Expert Concentration for Parameter-efficient Fine-tuning of Mixture-of-Expert LLMs"
_ICLR.cc/2026/Conference — Submitted to ICLR 2026_

### Official Review · Reviewer_9w4g · 2025-10-29

**Soundness:** 3
**Presentation:** 3
**Contribution:** 3
**Rating:** 4
**Confidence:** 4

**Summary:**

This paper observes that fine-tuning MoE models naturally concentrates domain knowledge into a small subset of experts, with routing scores increasingly favoring these specialists over training. Building on this insight, they design a lightweight two-stage method that first sharpens routing (attention + router tuning) then selectively updates only high-scoring experts based on their proposed Domain Advantage Score.

**Strengths:**

The core insight about expert concentration is valuable and well-supported empirically. The two-stage design follows naturally from this observation, and the method delivers strong results across different MoE architectures. The ablations effectively isolate the contribution of each component, and the analysis of expert distribution across layers adds useful interpretability.

**Weaknesses:**

(1)  The experiments mainly focus on mathematical reasoning and coding tasks, which are relatively structured domains with clear problem-solving patterns. It would strengthen the paper to evaluate the approach on more diverse tasks like open-ended generation, multilingual understanding, or knowledge-intensive QA. The current scope leaves open questions about whether the expert concentration phenomenon generalizes across different types of domain adaptation scenarios.

(2) While the Domain Advantage Score works well empirically, the paper doesn't deeply explore why this specific formula (difference of average routing scores) is optimal. Have the authors considered alternatives like KL divergence between domain and general distributions?

(3) All experiments are conducted on relatively smaller MoE models (2.7B to 16B parameters). Modern MoE models can be much larger like Qwen-30B-A3B or Qwen-80B-A3B. Does the expert concentration phenomenon hold at larger scales? Does the ratio of parameters to fine-tune (8%) remain constant, or should it be adjusted? Some discussion or preliminary results on scaling behavior would be valuable.

(4) The paper compares against FFT, LoRA, and ESFT, but recent work on MoE adaptation has proposed other methods. For example, there's no comparison with adapter-based approaches specifically designed for MoE, or more recent routing-aware PEFT methods.

**Questions:**

While the paper emphasizes parameter efficiency (8% trainable parameters), the actual training time and memory cost comparison is missing. The two-stage approach requires training twice and computing DAS scores, which adds overhead. Does this two-stage process actually save training time compared to methods that train fewer parameters in a single stage?

---

> ### Author Response · Authors · 2025-11-21
> **Response to the Concerns of Reviewer**
>
> We sincerely thank the reviewer for the valuable feedback. Below are detailed clarifications addressing the reviewers' concerns:
>
> ### **Response to Question 1: Computational Efficiency Comparation**
>
> To address this, we have added a dedicated comparison of average training time and peak GPU memory usage under the same training configuration. The results show that: Compared to FFT, our method achieves lower peak memory and faster training, because gradients and optimizer states are only allocated for a small subset of experts rather than for all parameters. Our method is also slightly more efficient than ESFT for fewer parameters tuning in the first stage. Moreover, we include a LoRA-based variant of our method, where LoRA adapters are applied only to the selected experts. This variant requires less peak memory and shorter training time than standard LoRA under the same setting, while also delivering better downstream performance.
>
> |                | Average Training Time(min) | Average Storage Spave(GB) | ACC   |
> | -------------- | -------------------------- | ------------------------- | ----- |
> | FFT            | 152                        | 42                        | 43.25 |
> | ESFT           | 120                        | 15                        | 40.55 |
> | LoRA           | 95                         | 5                         | 39.66 |
> | DAS-Tune       | 114                        | 12                       | 42.64 |
> | DAS-Tune(LoRA) | 88                         | 3.9                       | 40.14 |
>
> ### **Response to Weakness 2: Choice of Domain Advantage Metric**
>
> We thank the reviewer for this insightful question, and we have explicitly  experimented with three families of per-expert metrics in our prior experiments: (i) the difference of average routing scores between domain and general data (the DAS used in the main paper), (ii) a relative ratio of average routing scores (i.e., domain score divided by general score), and (iii) an expert-level KL score, defined as each expert’s contribution to the layer-wise KL divergence between the domain and general routing distributions. The comparative results of these three variants are reported in table below. Empirically, we observed that the relative ratio metric is less stable and consistently underperforms the other two, especially in layers where some experts have very small general-domain routing probabilities: in those cases, the ratio tends to amplify noise and over-emphasize experts with tiny absolute mass, which leads to worse expert selection and downstream performance. In contrast, the expert-level KL score performs very similarly to our difference-based DAS, both in terms of ranking the top domain experts and in final task performance. This is consistent with the fact that the KL term  $p \cdot log(p/q)$ depends on both the difference and the scale of probabilities, and in our setting the resulting rankings are highly correlated with those obtained from the raw difference of averages. Given its superior efficacy and simpler formulation, we ultimately adopted the absolute difference DAS for our measurement metric.
>
> |           | Gsm8k | MATH500 | MBPP  | Humaneval | Avg   |
> | --------- | ----- | ------- | ----- | --------- | ----- |
> |           | 43.38 | 10.80   | 40.80 | 30.48     | 35.45 |
> | DAS-Tune | 54.73 | 13.40   | 42.60 | 34.75     | 42.64 |
> | Rate      | 53.45 | 13.20   | 41.20 | 32.92     | 41.52 |
> | KL        | 54.51 | 13.20   | 42.60 | 34.14     | 42.44 |

---

> ### Author Response · Authors · 2025-11-21
> **Response to the Concerns of Reviewer**
>
> ### **Response to Weakness 3: Evaluation on Larger MoE Models**
>
> To further assess the generality and scalability of our approach, we conducted an additional set of experiments on a newer and larger sparse MoE model, Qwen3-30B-A3B. We first report the C-DAS values before and after attention tuning on Qwen3-30B-A3B and observe a similar increase, indicating that this phenomenon also persists in larger models.
>
> **The Dynamics of C-DAS**
> |      | Before Attention-Tuning | After Attention-Tuning |
> | ---- | ----------------------- | ---------------------- |
> | MATH | 0.59                    | 0.73                   |
> | CODE | 0.61                    | 0.73                   |
>
>
>
> We then select the domain experts based on their DAS scores and perform FFN-tuning process. The results below show that our method still achieves performance close to full-parameter fine-tuning while updating only about few model parameters, and it consistently outperforms the other baselines under the same training budget.
>
> **Qwen3-30B-A3B**
> |          | Gsm8k | MATH500 | MBPP  | Humaneval | Avg   |
> | -------- | ----- | ------- | ----- | --------- | ----- |
> |          | 85.37 | 57.80   | 49.60 | 51.21     | 70.35 |
> | FFT      | 87.11 | 68.60   | 58.80 | 60.97     | 75.96 |
> | ESFT     | 86.43 | 63.20   | 55.80 | 56.10     | 73.58 |
> | LoRA     | 85.29 | 60.80   | 53.80 | 53.65     | 71.92 |
> | DAS-Tune | 86.35 | 63.60   | 56.80 | 56.71     | 73.86 |
>
>
>
> We also performed an ablation on the number of tuned experts to examine how the fraction of trainable parameters affects performance on this 30B-scale model. Similar to our findings on the 16B model in the main paper, we observe that performance steadily improves as we increase the number of domain experts being fine-tuned, but the gains start to saturate once the proportion of trainable parameters exceeds roughly 12%. This pattern again indicates that it is sufficient to tune only a relatively small subset of experts in the MoE model to approach full fine-tuning performance, thereby achieving a favorable trade-off between training efficiency and downstream task accuracy.
>
> **Effect of Tuning Expert Count on Qwen3-30B-A3B**
> |               | Gsm8k | MATH500 | MBPP  | Humaneval | Avg   |
> | ------------- | ----- | ------- | ----- | --------- | ----- |
> |               | 85.37 | 57.80   | 49.60 | 51.21     | 70.35 |
> | DAS-Tune(4%)  | 85.37 | 61.20   | 53.60 | 53.04     | 71.96 |
> | DAS-Tune(8%)  | 86.35 | 63.60   | 56.80 | 56.71     | 73.86 |
> | DAS-Tune(12%) | 86.73 | 63.20   | 56.60 | 57.31     | 73.98 |
> | DAS-Tune(16%) | 86.66 | 64.60   | 56.80 | 58.53     | 74.34 |
>
>
> ### **Response to Weakness 1,4: Additional Dataset and  Baselines**
>
> We appreciate the reviewer’s suggestion to broaden the evaluation beyond mathematical reasoning and coding, we have added experiments on two knowledge-intensive subsets from MMLU,  physics and geography, to further validate the effectiveness of our proposed method, these new results (represented in table below) show that our approach continues to perform well in these settings. In terms of baselines, we have incorporated an additional routing-aware MoE PEFT method[1], this method first fine-tunes only the router parameters and then selects the experts with the highest post-tuning routing scores as the ones to be fine-tuned in the second stage. Under the same training budget and total number of trainable experts, our DAS-guided two-stage fine-tuning outperforms this router-lens baseline, suggesting that explicitly contrasting domain and general routing distributions leads to more reliable identification of truly domain-relevant experts. We also experimented with an adapter-style gate-tuning variant inspired by the GatePro [2], where an extra adapter is attached to the original router and initialized from pairwise cosine similarities between expert score vectors. However, this method was implemented in extremely large data regimes (on the order of 100B–500B tokens), and in our lower-data fine-tuning setting it did not yield competitive performance.
>
> |            | Gsm8k | MATH500 | MBPP  | Humaneval | Physics | Geography | Avg   |
> | ---------- | ----- | ------- | ----- | --------- | ------- | --------- | ----- |
> |            | 43.38 | 10.80   | 40.80 | 30.48     | 36.42   | 60.10     | 37.22 |
> | DAS-Tune   | 54.73 | 13.40   | 42.60 | 34.75     | 38.41   | 70.20     | 44.34 |
> | ESFT       | 52.46 | 13.20   | 39.00 | 32.92     | 37.74   | 68.69     | 42.37 |
> | FFT        | 55.34 | 15.00   | 42.60 | 34.15     | 39.07   | 72.22     | 45.05 |
> | LoRA       | 51.10 | 13.00   | 39.40 | 29.87     | 37.74   | 67.68     | 41.52 |
> | RouterLens | 52.24 | 13.00   | 39.60 | 31.71     | 37.09   | 69.19     | 42.26 |
>
> [1] Understanding and Leveraging the Expert Specialization of Context Faithfulness in Mixture-of-Experts LLMs
>
> [2] GatePro: Parameter-Free Expert Selection Optimization for Mixture-of-Experts Models

---

> > ### Comment · Reviewer_9w4g · 2025-11-27
> > **Response to Authors**
> >
> > Thank you for the detailed response. Most of my concerns are now addressed. However, I still have several concerns:
> >
> > 1. Scaling Performance: On the larger Qwen-30B model, paper's method (73.86) now underperforms Full Fine-Tuning (75.96). This looks inconsistent with main paper results where paper's method outperformed FFT on smaller models. This means that the method's advantage may diminish as model size increases.
> >
> > 2. Efficiency Trade-off: While faster than FFT, paper's method requires roughly 15GB of memory compared to LoRA's 5GB. Tripling the memory cost is a significant drawback for "parameter-efficient" tuning and make the claim of a "good balance" debatable.

---

> ### Author Response · Authors · 2025-11-27
> **Kindly Reminder for the Discussion**
>
> Dear Reviewer 9w4g,
>
> Thank you again for the time and insight you have devoted to evaluating our manuscript. We have tried our best to carefully address each of your comments in the rebuttal and clarified the points that were previously unclear.
>
> As the rebuttal period is drawing to a close, we would be grateful to know whether our responses have resolved your concerns, or if there are any additional questions we can clarify while time still permits. We are happy to provide further details and clarifications at your convenience.
>
> Thank you for your consideration, and we look forward to hearing from you.
>
> Best,
>
> The Authors

---

> ### Author Response · Authors · 2025-11-27
> **Response to the Concerns of Reviewer**
>
> We thank the reviewer again for the careful follow-up and constructive comments. We address the two remaining concerns below.
> ### **Response to Question 1: Scaling Performance**
>
> We realize that part of this concern comes from **a misunderstanding caused by our own presentation, and we apologize for the confusion. In the main results, our method does not consistently outperform full fine-tuning; rather, it is competitive with FFT and clearly better than other PEFT baselines**. The bold font in our tables was intended to highlight the best result among all non-FFT baselines, not to indicate that our method always surpasses FFT itself. The comment made us notice that the current caption is misleading on this point, and we will revise the caption.
>
> On the larger Qwen-30B model, the reported score  for our method is also close to FFT, and still stronger than the other PEFT baselines, which is consistent with the conclusions drawn from smaller models. To further address the scalability question, we have additionally run a more fine-grained ablation on Qwen-30B by varying the fraction of tuned experts. The results below show that tuning roughly 8–10% of parameters yields the best trade-off between performance and cost, and that this configuration remains competitive with FFT while clearly outperforming other baselines.  This optimal range of tuned-expert ratio is consistent with what we observed on the 16B model in the main paper, supporting our method's scalability.
>
> **Effect of Tuning Expert Count on Qwen3-30B-A3B**
> |               | Gsm8k | MATH500 | MBPP  | Humaneval | Avg   |
> | ------------- | ----- | ------- | ----- | --------- | ----- |
> |               | 85.37 | 57.80   | 49.60 | 51.21     | 70.35 |
> | DAS-Tune(4%)  | 85.37 | 61.20   | 53.60 | 53.04     | 71.96 |
> | DAS-Tune(8%)  | 86.35 | 63.60   | 56.80 | 56.71     | 73.86 |
> | DAS-Tune(10%) | 86.80 | 63.60   | 56.80 | 56.71     | 74.09 |
> | DAS-Tune(12%) | 86.73 | 63.20   | 56.60 | 57.31     | 73.98 |
> | DAS-Tune(14%) | 86.73 | 63.80   | 56.60 | 57.93     | 74.14 |
> | DAS-Tune(16%) | 86.66 | 64.60   | 56.80 | 58.53     | 74.34 |
>
>
> In addition, a key motivation of our method is not only peak performance on a single downstream task, but also better retention of the base model’s knowledge under continued training. To make this explicit, we have also conducted experiments where we continue training on another domain  corpus and then evaluate on the original benchmarks. The new table shows that, under the same number of continued-training steps, full fine-tuning suffers noticeably larger degradation on these base tasks, while our expert-tuning method exhibits substantially lower forgetting.
>
> **Continual Training Qwen3-30B-A3B using our Method**
> |           | Before | After | RPR   |
> | --------- | ------ | ----- | ----- |
> | GSM8K     | 86.35  | 83.93 | 0.972 |
> | MATH      | 63.60  | 61.60 | 0.969 |
> | MBPP      | 56.80  | 56.80 | 1.000 |
> | Humaneval | 56.71  | 55.49 | 0.978 |
> | Avg       | 73.86  | 72.09 | 0.976 |
>
> **Continual Training Qwen3-30B-A3B using FFT**
> |           | Before | After | RPR   |
> | --------- | ------ | ----- | ----- |
> | GSM8K     | 87.11  | 82.86 | 0.951 |
> | MATH      | 68.60  | 62.80 | 0.915 |
> | MBPP      | 58.80  | 58.80 | 1.000 |
> | Humaneval | 60.97  | 58.54 | 0.960 |
> | Avg       | 75.96  | 72.16 | 0.949 |

---

> ### Author Response · Authors · 2025-11-27
> **Response to the Concerns of Reviewer**
>
> ### **Response to Question 2: Efficiency Trade-off**
>
> We appreciate the concern about memory usage. Our goal here is to position the method as a middle ground between full fine-tuning and very lightweight PEFT methods: relative to full fine-tuning, our approach substantially reduces both the number of updated parameters and the training time, while achieving comparable or better performance; relative to LoRA, it does incur higher memory cost, but yields significantly stronger downstream results than LoRA and other PEFT baselines in our experiments.
>
> To more directly address the reviewer’s point, **we have also included results for a LoRA-style version of our method in the main results**. This variant applies our expert-selection mechanism but uses LoRA adapters on the selected experts instead of full expert tuning. As shown in the table below, this variant outperforms standard LoRA while using less memory (≈4 GB vs. 5 GB for LoRA) and shorter training time (≈88 minutes, which is faster than the LoRA baseline under the same setting). In other words, when memory is the primary constraint, practitioners can adopt this LoRA-based variant and obtain both better accuracy and lower resource usage than vanilla LoRA; when a moderate increase in memory is acceptable, the full expert-tuning variant offers further performance gains than full fine-tuning.
>
> |                | Average Training Time(min) | Average Storage Spave(GB) | ACC   |
> | -------------- | -------------------------- | ------------------------- | ----- |
> | FFT            | 152                        | 42                        | 43.25 |
> | ESFT           | 120                        | 15                        | 40.55 |
> | LoRA           | 95                         | 5                         | 39.66 |
> | DAS-Tune       | 114                        | 12                       | 42.64 |
> | DAS-Tune(LoRA) | 88                         | 3.9                       | 40.14 |

---

### Official Review · Reviewer_xuyw · 2025-10-29

**Soundness:** 3
**Presentation:** 3
**Contribution:** 2
**Rating:** 6
**Confidence:** 4

**Summary:**

The paper studies parameter-efficient fine-tuning (PEFT) for Mixture-of-Experts (MoE) language models and identifies instability caused by sparse expert activation. It introduces the Domain Advantage Score (DAS) to measure domain relevance of experts and observes an expert concentration phenomenon during fine-tuning. Based on this, a two-stage PEFT strategy is proposed: first tuning attention and router layers, then selectively fine-tuning domain-relevant experts. Extensive experiments on nine benchmarks show that the method achieves performance comparable to full fine-tuning while updating only a small fraction of parameters.

**Strengths:**

1. The paper presents a novel and well-motivated empirical finding that MoE models exhibit an expert concentration phenomenon, where domain-specific fine-tuning naturally focuses activation on a subset of experts, offering valuable insight into MoE behavior.

2. The proposed method creatively identifies domain-specific experts and applies a two-stage PEFT strategy—first tuning attention and router layers, then selectively fine-tuning expert modules—to achieve efficiency without sacrificing performance.

3. The experiments are extensive and well-designed, covering both domain-specific datasets and general benchmarks, effectively demonstrating the method’s strong performance and resistance to catastrophic forgetting.

4. The paper is well-structured and easy to follow.

**Weaknesses:**

1. The MoE base models used in the experiments are relatively outdated and all within a similar size range (around 16B parameters). Many newer MoE architectures of varying scales (e.g., Qwen3-30B-A3B, OLMoE-1B-7B) could be tested to provide a more comprehensive evaluation, and the lack of such experiments limits the generality and modern relevance of the findings.

2. In the Feature Analysis, all analyses are conducted on

**Questions:**

1. After Stage 1 training, certain experts are more frequently activated for specific domains. During Stage 1, did the authors include a load-balancing loss in the training objective? If so, could they provide an analysis of why load balancing fails to prevent expert over-activation? If not, can the authors demonstrate that the observed concentration of activated experts is not simply a consequence of omitting load balancing?

2. In Stage 2, the authors compute DAS values across all experts to rank their domain affinity and retain only the top-k experts. Why was this global top-k selection strategy chosen instead of retaining a fixed proportion of high-DAS experts within each layer?

3. Could the authors provide additional experimental results using newer or larger MoE models to further validate the generality and scalability of the proposed approach?

---

> ### Author Response · Authors · 2025-11-21
> **Response to the Concerns of Reviewer**
>
> We appreciate the thoughtful comments and concerns raised by the reviewer. We address each concern in detail below, highlighting the distinctions and contributions of our method.
>
> ### **Response to Question 1: Domain-Conditional Specialization Under Load Balancing**
>
> Our first-stage attention tuning incorporates a load-balancing loss, and the following explains why the C-DAS value increases during attention tuning:    In our setup, DAS for expert j at layer i is defined as the average routing score on the domain dataset minus that on the general dataset. the load-balancing objective operates on the global routing distribution over the mixed training data, while DAS  explicitly measure how differently experts are used on domain data versus general data.  Empirically, we observe that as we increase the number of attention-tuning steps, the per-expert average routing scores indeed become more evenly spread when aggregated over domain + general data, indicating that the load-balancing regularizer is effective at avoiding global expert collapse. However, at the conditional level, the router increasingly learns to route domain examples preferentially to a subset of experts while compensating by sending more general examples to the remaining experts. This makes the difference “domain minus general” larger for a few domain-relevant experts , so the sum of top-k DAS grows faster. Our results therefore do not indicate a failure of load balancing; rather, they show that under this soft global constraint, attention tuning naturally leads to more concentrated domain advantage.
>
> ### **Response to Question 2: Rationale Behind the Global Top-k Expert Selection Strategy**
>
> Our empirical observations indicate that the average DAS values vary substantially across layers, suggesting that different layers contribute unequally to domain specialization. In other words, some layers may exhibit stronger domain-specific expert differentiation, which is consistent with prior findings[1][2]. Motivated by this, we opted for a *global* selection scheme, which allows the model to flexibly allocate expert capacity to the layers where domain affinity is naturally concentrated, instead of enforcing an artificial per-layer quota that may over-allocate experts in less domain-sensitive layers and under-allocate in more crucial ones. To further validate this design choice, we conducted an additional comparison where we fixed the total number of fine-tuned experts but enforced a per-layer fixed-ratio selection. The results below show that the global top-k strategy consistently achieves better downstream performance under identical training budgets.
>
> |                | Gsm8k | MATH500 | MBPP  | Humaneval | Avg   |
> | -------------- | ----- | ------- | ----- | --------- | ----- |
> |                | 43.38 | 10.80   | 40.80 | 30.48     | 35.45 |
> | DAS-Tune(dyn)  | 54.73 | 13.40   | 42.60 | 34.75     | 42.64 |
> | DAS-Tune(topk) | 53.15 | 13.00   | 40.60 | 33.53     | 41.24 |
>
> [1] Let the Expert Stick to His Last: Expert-Specialized Fine-Tuning for Sparse Architectural Large Language Models
>
> [2] Domain-specific pruning of large mixture-of-experts models with few-shot demonstrations.

---

> ### Author Response · Authors · 2025-11-21
> **Response to the Concerns of Reviewer(2)**
>
> ### **Response to Question 3/Weakness 1: Evaluation on Larger MoE Models**
>
> To further assess the generality and scalability of our approach, we conducted an additional set of experiments on a newer and larger sparse MoE model, Qwen3-30B-A3B. We first report the C-DAS values before and after attention tuning on Qwen3-30B-A3B and observe a similar increase, indicating that this phenomenon also persists in larger models.
>
> **The Dynamics of C-DAS**
> |      | Before Attention-Tuning | After Attention-Tuning |
> | ---- | ----------------------- | ---------------------- |
> | MATH | 0.59                    | 0.73                   |
> | CODE | 0.61                    | 0.73                   |
>
>
>
> We then select the domain experts based on their DAS scores and perform FFN-tuning process. The results below show that our method still achieves performance close to full-parameter fine-tuning while updating only about few model parameters, and it consistently outperforms the other baselines under the same training budget.
>
> **Qwen3-30B-A3B**
> |          | Gsm8k | MATH500 | MBPP  | Humaneval | Avg   |
> | -------- | ----- | ------- | ----- | --------- | ----- |
> |          | 85.37 | 57.80   | 49.60 | 51.21     | 70.35 |
> | FFT      | 87.11 | 68.60   | 58.80 | 60.97     | 75.96 |
> | ESFT     | 86.43 | 63.20   | 55.80 | 56.10     | 73.58 |
> | LoRA     | 85.29 | 60.80   | 53.80 | 53.65     | 71.92 |
> | DAS-Tune | 86.35 | 63.60   | 56.80 | 56.71     | 73.86 |
>
>
>
> We also performed an ablation on the number of tuned experts to examine how the fraction of trainable parameters affects performance on this 30B-scale model. Similar to our findings on the 16B model in the main paper, we observe that performance steadily improves as we increase the number of domain experts being fine-tuned, but the gains start to saturate once the proportion of trainable parameters exceeds roughly 12%. This pattern again indicates that it is sufficient to tune only a relatively small subset of experts in the MoE model to approach full fine-tuning performance, thereby achieving a favorable trade-off between training efficiency and downstream task accuracy.
>
> **Effect of Tuning Expert Count on Qwen3-30B-A3B**
> |               | Gsm8k | MATH500 | MBPP  | Humaneval | Avg   |
> | ------------- | ----- | ------- | ----- | --------- | ----- |
> |               | 85.37 | 57.80   | 49.60 | 51.21     | 70.35 |
> | DAS-Tune(4%)  | 85.37 | 61.20   | 53.60 | 53.04     | 71.96 |
> | DAS-Tune(8%)  | 86.35 | 63.60   | 56.80 | 56.71     | 73.86 |
> | DAS-Tune(12%) | 86.73 | 63.20   | 56.60 | 57.31     | 73.98 |
> | DAS-Tune(16%) | 86.66 | 64.60   | 56.80 | 58.53     | 74.34 |

---

> ### Comment · Reviewer_xuyw · 2025-11-27
> **Response**
>
> I have read the authors' rebuttal and the other reviews. The authors have satisfactorily addressed my concerns. The addition experiments should be updated in the revision.  I will maintain my initial score.

---

### Official Review · Reviewer_KDPf · 2025-10-30

**Soundness:** 3
**Presentation:** 3
**Contribution:** 2
**Rating:** 4
**Confidence:** 4

**Summary:**

The paper proposes the Domain Advantage Score (DAS) metric to evaluate experts' relevance to their corresponding domains. Based on this metric, it introduces a lightweight two-stage PEFT framework: first adjusting attention and routers, then selectively fine-tuning expert modules. This achieves near-full-fledged accuracy using only a fraction of parameters.

**Strengths:**

1. The paper proposes the DAS metric and a two-stage PEFT framework to achieve fine-tuning results approaching full model accuracy.
2. Ablation experiments were conducted to validate the effectiveness of the DAS.
3. The model achieves results close to full fine-tuning by fine-tuning only 8% of its parameters.
4. Continuous learning demonstrates strong performance with minimal catastrophic forgetting.

**Weaknesses:**

1. DAS relies on differences in routing scores, yet these scores themselves are influenced by model initialisation, data bias, and noise, potentially lacking scalability.
2. The primary contribution of this paper lies in fine-tuning efficiency. Although only 8% of parameters are fine-tuned, this does not imply that only 8% of required GPU memory is needed. The entire model must still be loaded into GPU memory; savings are limited to the memory occupied by gradients and optimiser states. Therefore, I believe the authors should incorporate analyses of peak memory usage and training time or acceleration ratios.
3. The authors assert that their method maintains consistent training budgets with other approaches. Does this refer to matching the number of training steps in the second stage, or the total number of steps across both stages?
4. Experiments on continuous learning were conducted solely with the proposed method. Additional baselines may be required to demonstrate its advantages in continuous learning scenarios.

**Questions:**

1. Can the authors provide experimental analysis incorporating metrics such as peak memory usage, training duration, or acceleration ratios?
2. Can the authors elaborate on the specific allocation of training budget within the two-stage training process?
3. Can the authors furnish comparative experiments demonstrating continuous learning performance against other baselines?

---

> ### Author Response · Authors · 2025-11-21
> **Response to the Concerns of Reviewer**
>
> We appreciate the thoughtful comments and concerns raised by the reviewer. Below are detailed clarifications addressing the reviewers' concerns:
>
> ### **Response to Question 1/Weakness 2: Domain-Conditional Specialization Under Load BalancingComputational Efficiency Comparation**
>
> To address this, we have added a dedicated comparison of average training time and peak GPU memory usage  under the same training configuration in table below. The results show that: Compared to FFT, our method achieves lower peak memory and faster training, because gradients and optimizer states are only allocated for a small subset of experts rather than for all parameters. Our method is also slightly more efficient than ESFT for fewer parameters tuning in the first stage. Moreover, we include a LoRA-based variant of our method, where LoRA adapters are applied only to the selected experts. This variant requires less peak memory and shorter training time than standard LoRA under the same setting, while also delivering better downstream performance.
>
> |                | Average Training Time(min) | Average Storage Spave(GB) | ACC   |
> | -------------- | -------------------------- | ------------------------- | ----- |
> | FFT            | 152                        | 42                        | 43.25 |
> | ESFT           | 120                        | 15                        | 40.55 |
> | LoRA           | 95                         | 5                         | 39.66 |
> | DAS-Tune       | 114                        | 12                       | 42.64 |
> | DAS-Tune(LoRA) | 88                         | 3.9                       | 40.14 |
>
>
> ### **Response to Question 2/Weakness 3: Clarification on training budget and two-stage allocation**
>
> Our statement that the proposed method uses “consistent training budgets with other approaches” refers to matching the total number of training steps across both stages, rather than only the second stage. Concretely, all methods are trained with the same total budget of 1000 steps.
>
> For our two-stage procedure, this budget is allocated as follows:
>
> - **Stage 1 (attention tuning):** 500 steps
> - **Stage 2 (FFN /expert tuning):** 500 steps
>
> To further justify this choice, we conducted an additional ablation where we fixed the second-stage (FFN tuning) steps and varied the number of attention-tuning steps in Stage 1. The results (reported in Appendix B) show that:
>
> - Increasing the number of attention-tuning steps in Stage 1 generally improves performance,
> - but the gains exhibit diminishing returns beyond a certain point.
>
> This indicates that a limited number of steps in Stage 1 is sufficient to effectively steer the router and enhance the specialization of domain-relevant experts. Based on this observation, we set the Stage-1 budget to 500 steps as a practical trade-off between performance and efficiency, while keeping the total steps (1000) identical across all methods to ensure a fair comparison.
>
> ### **Response to Question 3/Weakness 4: Continuous learning baselines**
>
>  We agree that continuous learning performance should be evaluated against additional baselines. We have conducted  continuous learning experiments on other baselines(FFT, ESFT, LoRA). All methods share the same data stream, evaluation protocol, and hyperparameters as our method for a fair comparison. The results are presented in tables below. We observe that our method which achieves an RPR of 0.975 is higher than both ESFT and FFT, and on par with LoRA. FFT, which updates all experts, exhibits a noticeably lower RPR, indicating stronger interference and forgetting of previously acquired knowledge, suggesting that selectively tuning domain-relevant experts better stabilizes non-relevant experts and preserves prior knowledge. Although LoRA attains a similar RPR to ours, it updates far fewer parameters and consequently underperforms our method in final downstream task performance. This highlights that our approach provides a better trade-off between knowledge retention and task performance in MoE-based continuous learning.
>
> **ESFT**
> |           | Before | After | RPR   |
> | --------- | ------ | ----- | ----- |
> | GSM8K     | 52.46  | 50.56 | 0.964 |
> | MATH      | 13.20  | 13.00 | 0.970 |
> | MBPP      | 39.00  | 39.00 | 1.000 |
> | Humaneval | 32.92  | 31.70 | 0.963 |
> | Avg       | 40.55  | 39.42 | 0.972 |
>
> **FFT**
>
> |           | Before | After | RPR   |
> | --------- | ------ | ----- | ----- |
> | GSM8K     | 55.34  | 52.00 | 0.939 |
> | MATH      | 15.00  | 13.20 | 0.880 |
> | MBPP      | 42.60  | 41.00 | 0.962 |
> | Humaneval | 34.15  | 34.75 | 1.018 |
> | Avg       | 43.25  | 40.83 | 0.944 |
>
> **LoRA**
>
> |           | Before | After | RPR   |
> | --------- | ------ | ----- | ----- |
> | GSM8K     | 51.10  | 49.66 | 0.972 |
> | MATH      | 13.00  | 12.60 | 0.969 |
> | MBPP      | 39.40  | 39.00 | 0.990 |
> | Humaneval | 29.87  | 29.27 | 0.980 |
> | Avg       | 39.66  | 38.70 | 0.975 |

---

> ### Author Response · Authors · 2025-11-23
> **Response to the Concerns of Reviewer**
>
> ### **Response to Weakness 1: Evaluation on Larger MoE Models**
>
> To further assess the generality and scalability of our approach, we conducted an additional set of experiments on a newer and larger sparse MoE model, Qwen3-30B-A3B. We first report the C-DAS values before and after attention tuning on Qwen3-30B-A3B and observe a similar increase, indicating that this phenomenon also persists in larger models.
>
> **The Dynamics of C-DAS**
> |      | Before Attention-Tuning | After Attention-Tuning |
> | ---- | ----------------------- | ---------------------- |
> | MATH | 0.59                    | 0.73                   |
> | CODE | 0.61                    | 0.73                   |
>
>
>
> We then select the domain experts based on their DAS scores and perform FFN-tuning process. The results below show that our method still achieves performance close to full-parameter fine-tuning while updating only about few model parameters, and it consistently outperforms the other baselines under the same training budget.
>
> **Qwen3-30B-A3B**
> |          | Gsm8k | MATH500 | MBPP  | Humaneval | Avg   |
> | -------- | ----- | ------- | ----- | --------- | ----- |
> |          | 85.37 | 57.80   | 49.60 | 51.21     | 70.35 |
> | FFT      | 87.11 | 68.60   | 58.80 | 60.97     | 75.96 |
> | ESFT     | 86.43 | 63.20   | 55.80 | 56.10     | 73.58 |
> | LoRA     | 85.29 | 60.80   | 53.80 | 53.65     | 71.92 |
> | DAS-Tune | 86.35 | 63.60   | 56.80 | 56.71     | 73.86 |
>
>
>
> We also performed an ablation on the number of tuned experts to examine how the fraction of trainable parameters affects performance on this 30B-scale model. Similar to our findings on the 16B model in the main paper, we observe that performance steadily improves as we increase the number of domain experts being fine-tuned, but the gains start to saturate once the proportion of trainable parameters exceeds roughly 12%. This pattern again indicates that it is sufficient to tune only a relatively small subset of experts in the MoE model to approach full fine-tuning performance, thereby achieving a favorable trade-off between training efficiency and downstream task accuracy.
>
> **Effect of Tuning Expert Count on Qwen3-30B-A3B**
> |               | Gsm8k | MATH500 | MBPP  | Humaneval | Avg   |
> | ------------- | ----- | ------- | ----- | --------- | ----- |
> |               | 85.37 | 57.80   | 49.60 | 51.21     | 70.35 |
> | DAS-Tune(4%)  | 85.37 | 61.20   | 53.60 | 53.04     | 71.96 |
> | DAS-Tune(8%)  | 86.35 | 63.60   | 56.80 | 56.71     | 73.86 |
> | DAS-Tune(12%) | 86.73 | 63.20   | 56.60 | 57.31     | 73.98 |
> | DAS-Tune(16%) | 86.66 | 64.60   | 56.80 | 58.53     | 74.34 |

---

> ### Author Response · Authors · 2025-11-27
> **Kindly Reminder for the Discussion**
>
> Dear Reviewer KDPf,
>
> Thank you again for the time and insight you have devoted to evaluating our manuscript. We have tried our best to carefully address each of your comments in the rebuttal and clarified the points that were previously unclear.
>
> As the rebuttal period is drawing to a close, we would be grateful to know whether our responses have resolved your concerns, or if there are any additional questions we can clarify while time still permits. We are happy to provide further details and clarifications at your convenience.
>
> Thank you for your consideration, and we look forward to hearing from you.
>
> Best,
>
> The Authors

---

### Official Review · Reviewer_RKrV · 2025-10-31

**Soundness:** 3
**Presentation:** 3
**Contribution:** 3
**Rating:** 8
**Confidence:** 3

**Summary:**

This paper addresses the problem of parameter-efficient fine-tuning (PEFT) for Mixture-of-Experts (MoE) large language models. The authors argue that directly applying PEFT methods designed for dense models to MoE architectures is suboptimal due to training instability caused by sparse expert activation. To tackle this, the paper first conducts an empirical study, revealing an expert concentration phenomenon during domain-specific fine-tuning, where computation for a specific domain gradually concentrates on a small subset of experts. Building on this finding, the authors propose a simple yet effective metric, the Domain Advantage Score(DAS), to quantify and identify these domain-relevant experts. Subsequently, they introduce a lightweight two-stage fine-tuning framework. Experiments conducted on multiple math and coding benchmarks demonstrate that this approach achieves performance comparable to full fine-tuning with only about 8% of the trainable parameters. Furthermore, it effectively mitigates catastrophic forgetting of the model's general capabilities.

**Strengths:**

1.Solid Empirical Analysis: The paper's primary strength lies in its in-depth investigation of MoE fine-tuning dynamics. Instead of merely proposing a new method, the authors first uncover the core phenomenon of "expert concentration" through empirical study and design the DAS metric to quantify it. This foundational analysis provides a strong theoretical and experimental basis for the proposed method.
2.Comprehensive and Convincing Experiments: The empirical evaluation is exceptionally thorough, including ablation study, variation study and so on.
3.Clarity and Presentation: The paper is well-structured, clearly written, and easy to follow.

**Weaknesses:**

1. The calculation of DAS relies on a "general dataset" (Dg) to contrast with the target domain data. The paper does not elaborate on the selection criteria for this dataset or the sensitivity of the method to its choice.

**Questions:**

1. How was the general dataset (Dg) selected for the experiments? Is the performance of the method sensitive to the choice of Dg?

---

> ### Author Response · Authors · 2025-11-21
> **Response to the Concerns of Reviewer**
>
> Thank you for your valuable question. We clarify the construction process and sensitivity analysis of Dg as follows:
>
> ### **Response to Question 1/Weakness 1: How we selected and constructed Dg**
>
>  To approximate a general commonsense reasoning distribution, we combined two representative benchmark datasets — ARC and CommonsenseQA — which cover diverse, real-world, non-domain-specific reasoning scenarios. We then used a strong LLM to generate chain-of-thought style reasoning for each question and applied a two-stage filtering process: we discarded instances where the final answer was incorrect, and we further removed examples whose reasoning chains were self-contradictory or extremely short. Importantly, because ARC and CommonsenseQA do not focus on the specific domains we adapt to, Dg serves as a genuinely “general” reference distribution against which domain-specific routing patterns can be contrasted.
>
> ### **Response to Question 1/Weakness 1: Sensitivity analysis regarding the choice of Dg**
>
>  To evaluate whether our method depends heavily on the specific sample choice within Dg, we performed three independent random samplings following the same pipeline described in the paper. The detected domain-expert groups remained highly stable across trials, with an overlap ratio greater than 95%, and the induced rankings of high-DAS experts were nearly identical. At the level of downstream performance, the variance of fine-tuning results across these three runs was within ±0.5%, indicating that small perturbations in the composition of Dg do not materially affect either which experts are picked or how well the method performs.
>
> |           | Gsm8k | MATH500 | MBPP  | Humaneval | Avg   |
> | --------- | ----- | ------- | ----- | --------- | ----- |
> |           | 43.38 | 10.80   | 40.80 | 30.48     | 35.45 |
> | DAS-Tune1 | 54.73 | 13.40   | 42.60 | 34.75     | 42.64 |
> | DAS-Tune2 | 54.59 | 13.60   | 42.00 | 34.75     | 42.49 |
> | DAS-Tune3 | 54.20 | 13.40   | 42.00 | 33.53     | 42.16 |
> | ESFT      | 52.46 | 13.20   | 39.00 | 32.92     | 40.55 |
> | FFT       | 55.34 | 15.00   | 42.60 | 34.15     | 43.25 |

---

### Author Response · Authors · 2025-12-03
**Authors' Official Summary on Submission 24856**

Dear Area Chair,

Due to the special circumstances of this year's review process, we have prepared a summary of our rebuttal for your convenience. It demonstrates how our revisions address the reviewers' concerns.

### **Reviewer's feedback**

Reviewers highlighted the paper's strength in uncovering the domain expert concentration phenomenon in MoE models and quantifying this using the Domain Advantage Score (DAS) metric (RKrV, xuyw, 9w4g). They emphasized that this empirical analysis provides a conceptual and experimental basis for the proposed methodology (RKrV, xuyw). Reviewers appreciated the resulting two-stage PEFT framework for achieving performance approaching that of full fine-tuning while updating only approximately 8% of parameters (KDPf, xuyw, 9w4g), with continuous learning experiments further demonstrating strong performance and minimal catastrophic forgetting (RKrV, KDPf, xuyw).

### **Main Concerns of Reviewers**

Here we summarize the reviewers' main concerns and detail how we addressed them in this rebuttal：

**Computational Efficiency Comparation(Reviewer KDPf/9w4g)**

To address this, we have added a dedicated comparison of average training time and peak GPU memory usage under the same training configuration. The results show that: Compared to FFT, our method achieves lower peak memory and faster training. Our method is also slightly more efficient than ESFT for fewer parameters tuning in the first stage. Moreover, the LoRA-based variant of our method requires less peak memory and shorter training time than standard LoRA under the same setting, while also delivering better downstream performance.

**Evaluation on Larger MoE Models(Reviewer KDPf/xuyw/9w4g)**

To further assess the scalability of our approach, we conducted an additional set of experiments on a larger sparse MoE model, Qwen3-30B-A3B. We first report the C-DAS values before and after attention tuning on Qwen3-30B-A3B and observe a similar increase, indicating that this phenomenon also persists in larger models. We then select the domain experts based on their DAS scores and perform FFN-tuning process. The results below show that our method still achieves performance close to full-parameter fine-tuning while updating only about few model parameters, and it consistently outperforms the other baselines under the same training budget. We also performed an ablation on the number of tuned experts to examine how the fraction of trainable parameters affects performance. The results show that tuning roughly 8–10% of parameters yields the best trade-off between performance and cost, which is consistent with what we observed in the main paper.

**Clarification on Training Budget and Two-stage Allocation(Reviewer KDPf)**

Our statement that the proposed method uses “consistent training budgets with other approaches” refers to matching the total number of training steps across both stages, rather than only the second stage. Concretely, all methods are trained with the same total budget of 1000 steps to ensure a fair comparison of results.

**Continuous Learning Baselines(Reviewer KDPf)**

We have conducted continuous learning experiments on other baselines(FFT, ESFT, LoRA). It shows that our method which achieves an RPR of 0.975 is higher than both ESFT and FFT, and on par with LoRA. FFT, which updates all experts, exhibits a noticeably lower RPR, indicating stronger interference and forgetting of previously acquired knowledge, suggesting that selectively tuning domain-relevant experts better preserves prior knowledge. Although LoRA attains a similar RPR to ours, it updates far fewer parameters and consequently underperforms our method in final downstream task performance.

**Choice of Domain Advantage Metric and Global Top-k Expert Selection Strategy(Reviewer xuyw/9w4g)**

For Domain Advantage Metric, we have explicitly experimented with several per-expert metrics(absolute difference/relative ratio/expert-level KL score). The results demonstrated that our proposed method yielded the best performance.

For Global Top-k Expert Selection Strategy, our empirical observations indicate that the average DAS values vary substantially across layers, suggesting that different layers contribute unequally to domain specialization. In other words, some layers may exhibit stronger domain-specific expert differentiation, which is consistent with prior findings[1][2]. To further validate this design choice, we conducted an additional comparison where we fixed the total number of fine-tuned experts but enforced a per-layer fixed-ratio selection. The results below show that the global top-k strategy consistently achieves better downstream performance under identical training budgets.

---

> ### Author Response · Authors · 2025-12-03
> **Authors' Official Summary on Submission 24856(2)**
>
> **Additional Dataset and Baselines(Reviewer 9w4g)**
>
> To broaden the evaluation beyond mathematical reasoning and coding, we have added experiments on two knowledge-intensive subsets from MMLU, physics and geography. We also add additional baselines to further validate the effectiveness of our proposed method, and the results show that our approach continues to perform well in these settings.
>
> **How we constructed General Dataset and Sensitivity Analysis(Reviewer RKrV)**
>
> We detail the construction of the General Dataset in responses and present a corresponding sensitivity analysis. Specifically, we performed three independent random samplings following the same pipeline described in the paper. The detected domain-expert groups remained highly stable across these trials and the variance of fine-tuning results across these three runs was within 0.5%, indicating that small perturbations in the composition of Dg do not materially affect either which experts are picked or how well the method performs.
>
>
> ### **Rebuttal Response Status**
>
> Among the reviewers who have responded to us, Reviewer xuyw indicated that their concerns have been satisfactorily resolved. Reviewer 9w4g also stated that most of their concerns were addressed. We provided further explanation regarding the two remaining concerns, which stemmed from misunderstandings of the method. However, the subsequent system bug prevented Reviewer 9w4g from continuing the discussion with us.
>
> We hope the above summary will help the Reviewers, Area Chairs, and Senior Area Chairs assess the overall scope of the work and comments, navigate through the discussion, and make an objective and unbiased assessment. Thank you for your valuable time.
>
> Best,
>
> The Authors
>
>
> [1] Let the Expert Stick to His Last: Expert-Specialized Fine-Tuning for Sparse Architectural Large Language Models
>
> [2] Domain-specific pruning of large mixture-of-experts models with few-shot demonstrations.

---

### Meta-Review · Area_Chair_QmrH · 2026-01-08

**Summary:**

Reviewers are split 2-2, with supporters noting the novel expert concentration finding and critics raising concerns about limited experimental scope (outdated ~16B models, only math/coding tasks, missing efficiency metrics). The critics' concerns are well-supported since modern MoE research has moved to larger architectures (Qwen3, DeepSeek-V3), and the limited setup may contain confounding variables. Overall, I recommend rejection.

**Reviewer Concerns:**

see above

**Reviewer Scores:**

Discussion was sufficient; the split in scores concerns experimental scope which cannot be resolved without additional experiments, and scores would have remained similar.

---

### Decision · Program_Chairs · 2026-01-26

Reject